# Fluorescent Pyranoindole Congeners: Synthesis and Photophysical Properties of Pyrano[3,2-*f*], [2,3-*g*], [2,3-*f*], and [2,3-*e*]Indoles

**DOI:** 10.3390/molecules27248867

**Published:** 2022-12-13

**Authors:** Ainur D. Sharapov, Ramil F. Fatykhov, Igor A. Khalymbadzha, Maria I. Valieva, Igor L. Nikonov, Olga S. Taniya, Dmitry S. Kopchuk, Grigory V. Zyryanov, Anastasya P. Potapova, Alexander S. Novikov, Vladimir V. Sharutin, Oleg N. Chupakhin

**Affiliations:** 1Department of Organic and Biomolecular Chemistry, Ural Federal University, Mira Street 19, 620002 Ekaterinburg, Russia; 2Institute of Organic Synthesis, Ural Branch of the Russian Academy of Sciences, Kovalevskoy Street 22, 620219 Ekaterinburg, Russia; 3Institute of Chemistry, Saint Petersburg State University, Universitetskaya Emb., 7/9, 199034 Saint Petersburg, Russia; 4Research Institute of Chemistry, Peoples’ Friendship University of Russia (RUDN University), Miklukho-Maklaya Street, 6, 117198 Moscow, Russia; 5Department of Chemistry, Institute of Natural Sciences, South Ural State University (National Research University), Lenin Avenue 76, 454080 Chelyabinsk, Russia

**Keywords:** pyranoindole, Bischler–Möhlau reaction, Nenitzescu reaction, Pechmann condensation, Stokes shift, Lippert-Mataga equation

## Abstract

This paper reports the synthesis of four types of annulated pyranoindole congeners: pyrano[3,2-*f*]indole, pyrano[2,3-*g*]indole, pyrano[2,3-*f*]indole, and pyrano[2,3-*e*]indole and photophysical studies in this series. The synthesis of pyrano[3,2-*f*], [2,3-*g*], and [2,3-*e*]indoles involve a tandem of Bischler–Möhlau reaction of 3-aminophenol with benzoin to form 6-hydroxy- or 4-hydroxyindole followed by Pechmann condensation of these hydroxyindoles with β-ketoesters. Pyrano[2,3-*f*]indoles were synthesized through the Nenitzescu reaction of *p*-benzoquinone and ethyl aminocrotonates and subsequent Pechmann condensation of the obtained 5-hydroxyindole derivatives. Among the pyranoindoles studied, the most promising were pyrano[3,2-*f*] and [2,3-*g*]indoles. These compounds were characterized by moderate to high quantum yields (30–89%) and a large (9000–15,000 cm^−1^) Stokes shift. More detailed photophysical studies were carried out for a series of the most promising derivatives of pyrano[3,2-*f*] and [2,3-*g*]indoles to demonstrate their positive solvatochromism, and the data collected was analyzed using Lippert-Mataga equation. Quantum chemical calculations were performed to deepen the knowledge of the absorption and emission properties of pyrano[3,2-*f*] and [2,3-*g*]indoles as well as to explain their unusual geometries and electronic structures.

## 1. Introduction

Pyranocarbazoles, benzoderivatives of pyrano[2,3-*g*]indole, represent a structural motif that is ubiquitous among compounds with luminescence properties. Many pyranocarbazole derivatives have been described, and their spectral characteristics have been studied [1]. Moreover, the outstanding photophysical properties of this series of compounds determine the range of their possible applications: for example, pyranoindoles are known as components of phototriggered drug delivery systems for the photocontrolled release of chlorambucil [2], a two-photon active photoinitiator for 3D stereolithography [3], a two-photon fluorescent probe for imaging carbon monoxide in living tissues [4], host materials for blue, green, and red phosphorescent organic light-emitting diodes [5], a one- and two-photon activated photoremovable protecting groups [6], and a photo-cross-linker for cross-linking pyrimidines in nucleic acids under visible light irradiation [7,8].

Pyranoindoles can be considered as simplified analogues of pyranocarbazoles, which contain one less benzene ring and, as a consequence, are more soluble in polar solvents (e.g., physiological fluids) [9,10]. Pyranoindoles are known to have anti-inflammatory [11], fibrinolytic, antiulcer, antidepressant, analgesic, and antiproliferative activities (Figure 1) [12]. However, publications on the photophysical properties of pyranoindoles are very limited. Only a few reports of the photophysical characteristics of pyranoindoles have been published to date [13,14]. Among them, only one article was devoted to a detailed study of the photophysical characteristics (quantum yield, absorption and emission maxima, etc.) of pyrano[2,3-*f*] and [3,2-*e*]indoles [15]. Leader compounds obtained in this previous work were characterized by quantum yields of 0.48–0.57 and Stokes shifts of 109–120 nm. Large Stokes shifts can be practically useful, for example, for optoelectronic applications [16] and in biological applications, especially fluorescent microscopy. The large Stokes shift, which is usually over 150 nm, is particularly useful for super-resolution optical microscopy of biological objects [17,18], Förster resonance energy transfer (FRET) microscopy [19], and possible applications in photoelectronic devices, as non-linear materials, emitters, or in organic light-emitting diodes (OLED) [20].

Thus, the aim of this work was the synthesis and systematic study of the photophysical characteristics of four different types of pyranoindoles with [2,3-*f*], [2,3-e], [3,2-*f*], and [2,3-g] type of ring fusion.

## 2. Results

### 2.1. Synthesis

#### 2.1.1. Synthesis of Pyrano[2,3-*f*]indoles

Fluorophores with pyrano[2,3-*f*]indole core have previously been studied [15], and we began our study from pyranoindoles with this [2,3-*f*] ring fusion type.

Pechmann condensation of 5-hydroxyindole **1** with 2-ethoxycarbonylcyclohexanone in the presence of methanesulfonic acid (MsOH) afforded pyranocoumarin **2** in a 61% yield (Figure 1). The signals of hydrogen atoms at positions 7 and 11 confirmed the [2,3-*f*]-type of fusion between pyrane and indole rings (Figure 1). These signals were registered as singlets at 7.6 and 7.4 ppm, ruling out an alternative pyrano[3,2-*e*]indole structure characterized by doublets with a spin-spin coupling (SSC) constant of 8.8 Hz [15,21,22]. Compound **2** was converted to the corresponding 8-unsubstituted indole **3** upon refluxing in acetic acid with the addition of sulfuric acid (Figure 1).

Although compound **2** has moderate photophysical characteristics (see Section 2.2), pyrano[2,3-*f*]indole core almost completely lost its fluorescent properties upon decarboxylation. Therefore, we continued our search for photoactive compounds among pyranoindoles with other types of fusion of the pyran and indole rings.

#### 2.1.2. Synthesis of Pyrano[2,3-*e*]indoles

In the Pechmann condensation of 3-aminophenol with β-ketoesters, 4-hydroxy and 6-hydroxyindoles served as convenient starting materials for the synthesis of three isomeric pyranoindoles–[2,3-*e*], [3,2-*f*], and [2,3-*g*]. These hydroxyindoles can be easily obtained in one step using the Bischler–Möhlau reaction [21,22]. A condensation of 3-aminophenol with benzoin (Figure 2) provided a mixture of 4-hydroxy- and 6-hydroxy-2,3-diphenylindoles **4** and **5**, respectively, in 78% overall yield [21,22]. The indoles **4** and **5** were separated by silica gel column chromatography using CH_2_Cl_2_-hexane (1:1) as the eluent.

The Pechmann condensation of 4-hydroxyindole **4** with ethyl benzoyl acetate or ethyl 3-oxo-hexanoate resulted in pyrano[2,3-*e*]indoles **6a** and **6b** in 68 and 63% yield, respectively (Figure 3). NMR spectral data are consistent with the ascribed structure of indoles **6a** and **6b**. In particular, they are characterized by the SSC constant of 8.7 Hz between the protons at positions 5 and 6 [23]. To make sure that the structure assigned on the basis of NMR spectroscopy is correct, we also proved the structure of compound **6b** using X-ray diffraction analysis (Figure 2).

4-Propylpyranoindole **6b** was characterized by a quantum yield of luminescence of 7%, while its phenyl counterpart **6a** did not exhibit luminescent properties at all (see Section 2.2). Thus, indoles **6** have weak fluorescent properties, and this direction was also abandoned.

#### 2.1.3. Synthesis of Pyrano[3,2-*f*] and [2,3-*g*]Indoles

Unlike 4-hydroxyindole, 6-Hydroxyindole contains two positions of the benzene cycle that can participate in the Pechmann reaction (C5 and C7 atoms), so when it reacts with β-ketoesters one can expect the formation of two isomeric pyranoindoles with linear [3,2-*f*] and angular [2,3-*g*] type of ring fusion.

In accordance with this assumption, the interaction of the acetoacetic ester with 6-hydroxyindole yielded a mixture of isomeric linear and angular pyranoindoles **7a** and **8a**, respectively (Figure 4). The mixture of compounds **7a** and **8a** was separated using silica gel column chromatography. The structures of the obtained compounds were established on the basis of NMR spectroscopy. Angular pyrano[2,3-*g*]indole **8a** is characterized by doublets of C4H and C5H protons at 7.2 and 7.7 ppm with an SSC constant of 8.6 Hz, whereas linear pyrano[3,2-*f*]indole **7a** has no SSC between protons at C5 and C9 (Figure 4). Compounds **7a** and **8a** showed acceptable photophysical properties (quantum yields of luminescence of 14 and 58%, respectively, and an emission maxima at 500 nm and a Stokes shift of 167–209 nm) and therefore were chosen as a starting point for further work. The reactions of hydroxyindole **5** with 2-methylacetoacetic ester, ethyl 3-oxohexanoate, 2-benzyl acetoacetic ester, and 2-chloroacetoacetic ester provided corresponding pairs of pyranoindole regioisomers **7b**–**e** and **8b**–**e**. In addition, pyranoindoles **7f** and **8f** and **7g** and **8g** were obtained via the reaction of hydroxyindole **5** with cyclic ketoesters. It is interesting to note that in the case of the Pechmann reaction with ethyl cycloheptanone-2-carboxylate and ethyl benzoylacetate (Figure 4), no angular pyranoindoles **8h** and **8i** were detected, and only linear isomers **7h** and **7i** were obtained from the reaction mixture. We attributed the formation of the single isomer to unfavorable steric conditions for cyclization at position 7 of hydroxyindole **5**.

#### 2.1.4. Modification of Pyrano[3,2-*f*] and [2,3-*g*]Indoles

It is known that alkylation of carbazoles can affect their photophysical properties and solubility [24]. Therefore, we alkylated the most promising pyranoindoles **7c**,**g**,**i** in attempts to modify their photophysical properties. The *N*-ethyl and *N*-benzyl derivatives **9a**–**d** were obtained by the reaction of pyranoindoles with ethyl iodide or benzyl bromide. A high-yield reaction occurred when indole reacted with alkylating agents in the presence of sodium hydride in DMF (Figure 5).

Annulation of a benzene ring to a fluorophore system is known to improve its photophysical characteristics [25,26,27]. Therefore, we performed aromatization of the cyclohexene moiety to a benzene fragment in compound **8g** (Figure 6). The reaction was carried out with 3 equiv. of DDQ in refluxing dichloroethane, providing benzoderivative **10**.

Finally, we introduced substituents in the phenyl groups of the indole ring to estimate the influence of these groups on the photophysical characteristics. For this purpose, we used condensation of hexamethoxybenzoin with 3-aminophenol, yielding 6-hydroxyindole **11**. This indole was involved in the Bischler–Möhlau reaction leading to pyranocoumarin **12** (Figure 7).

### 2.2. Photophysical Studies

#### 2.2.1. Absorption and Emission in Acetonitrile

To compile the photophysical profile of pyranoindoles a pair of pyrano[2,3-*f*]indoles derivatives **2**–**3**, ten isomeric pyranoindoles with linear [3,2-*f*]- or angular [2,3-*g*]-type of ring fusion, such as **7a**–**g**, **8a**–**g**, **7h**, **7i**, **10** and **12**, N-ethyl and N-benzyl derivatives of pyrano[3,2-*f*]- or [2,3-*g*]indoles **9a**–**d**, and, finally, a pair of pyrano[2,3-*e*]indoles **6a**–**b** were studied. In addition, structure-property correlation studies of donor-acceptor (D-A) fused-heterocyclic fluorophores involving simple separated motifs of pyranoindoles **13**–**15** were performed.

All fluorophores were soluble at concentrations below 2 × 10^−5^ M both in non-polar solvents (n-heptane, toluene), weakly and strongly polar aprotic solvents (THF, dichloromethane, acetonitrile, DMF, DMSO), and in the highly polar protic solvent (methanol). All compounds fluoresce in solution.

The photophysical properties of pyranoindole fluorophores in acetonitrile were collected in Table 1.

**Table 1 molecules-27-08867-t001:** Experimental photophysical data of the pyranoindole compounds and their structural motifs in acetonitrile at room temperature.

Compound	Structure	λ_abs_, nm (εM (10^4^ M^−1^ cm^−1^)) ^a^	λ_em_, nm ^b^	Stokes Shift, cm^−1^ (nm)	φ, % ^c^
**2**	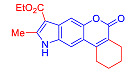	271 (sh) (2.9), 329 (4.1)	416	6357 (145)	24
**3**	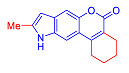	274 (0.6)	447	6868 (173)	<0.1
**6a**	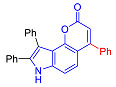	241 (4.4), 285 (1.0)	522	15,931 (281)	0.1
**6b**	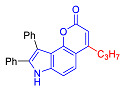	241 (3.3), 280 (1.6)	474	14,617 (233)	7
**7a**	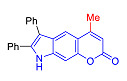	335 (0.8)	502	9930 (167)	14
**8a**	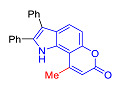	287 (1.1)	496	14,682 (209)	57
**7b**	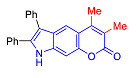	284 (1.4), 337 (0.7)	491	9307 (207)	7
**8b**	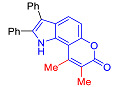	275 (0.7), 298 (0.6)	481	12,767 (206)	49
**7c**	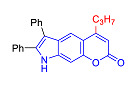	271 (sh) (0.7), 300 (0.5)	499	13,293 (199)	89
**8c**	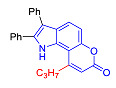	268 (0.6), 300 (0.6)	494	13,090 (194)	76
**7d**	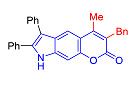	338 (3.3)	500	9586 (162)	10
**8d**	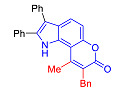	278 (2.7), 306 (2.8)	489	12,230 (183)	66
**7e**	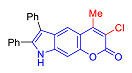	225 (1.6), 303 (1.4), 330 (sh) (1.1)	420, 520	10,618 (217)	4
**8e**	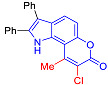	243 (1.4) 305 (1.1)	416, 524	8025 (215)	33
**7f**	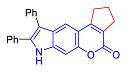	275 (5.6), 337 (0.9)	490	9265 (215)	11
**8f**	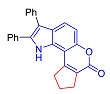	271 (2.9), 302 (2.7)	473	12,872 (171)	60
**7g**	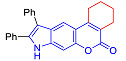	334 (3.5)	471	8709 (137)	25
**8g**	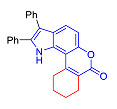	274 (4.3), 302 (4.7)	475	12,060 (173)	61
**10**	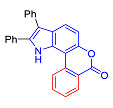	268 (2.8), 322 (1.6)	516	11,676 (248)	6
**7h**	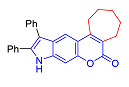	275 (3.3), 337 (0.8)	421	5921 (146)	15
**7i**	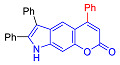	245 (2.5), 303 (2.4), 335 (sh) (1.7)	500	12,680 (197)	9
**12**	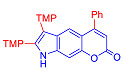	315 (1.1)	593	14,883 (278)	<0.1
**13**	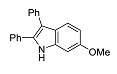	226 (2.8), 325 (1.7)	418	7036 (157)	98
**14**	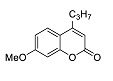	241 (3.3), 279 (2.9), 307 sh (2.3)	375	5907 (134)	2
**15** [28]	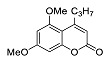	317	393	6100 (76)	<0.1
**9a**	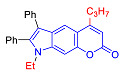	285 (1.3), 335 (0.7)	522	10,694 (281)	9
**9b**	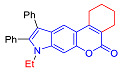	284 (2.2), 329 (1.5)	420	6586 (136)	10
**9c**	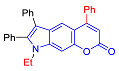	341 (0.9)	586	12,261 (245)	<0.1
**9d**	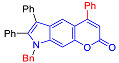	274 (3.4), 339 (0.8)	570	11,955 (231)	<0.1

^a^ Absorption spectra from 230 to 450 nm were measured at 20 °C in MeCN, ^b^ Emission spectra were measured at r.t. in MeCN, ^c^ Absolute quantum yields were measured at r.t. in MeCN using integrating sphere of the Horiba-Fluoromax-4 [29].

For the most significant pyranoindole derivatives **7**–**9**, normalized absorption (Figure 3) and emission (Figure 4) spectra graphs were plotted, which allowed for a comparative analysis of the obtained spectroscopic data.

The absorption spectra of the analyzed fluorophores represent two bands of different intensities with maxima in the wavelength ranges of 275–325 nm and 330–400 nm, which correspond to the S0 → S2 and S0 → S1 transitions (Figure 3). The experimental absorption bands correlate well with the absorption spectra calculated by the CAM-B3LYP/6-31+G* method in the gas phase for the n-propyl substituted isomers **7c**–**8c** (Appendix A). It can be seen from the experimental data that the intensity of each absorption band for compounds **7**–**9** depends on the nature of the functional groups at positions 3–4 for linear isomers and 8–9 for angular isomers. Thus, the cycloalkane-fused derivatives **7h** and **7f** demonstrate the dominant band due to the S0 → S2 transition with εM > 55,000 M^−1^ cm^−1^, while the absorption bands for the fluorophores **7b**–**8b**, **7c**–**8c**, **7e**–**8e**, **8f**, and **9a** represent a balance between two transitions (S0 → S2 and S0 → S1) with εM < 10,000 M^−1^.

The emission spectra of fluorophores **7b**–**8b**, **7c**–**8c**, **7f**–**8f**, **7h**, and **9a** were observed as continuous unstructured emission bands with a maxima from 421 to 522 nm, related to the excited ICT state in a highly polar aprotic solvent, which was confirmed by DFT calculations (see Figure 3 and Figure 4).

In the case of 3- and 8-chloro substituted chromophores **7e**–**8e**, a dual (hybrid) emission was observed [30]. The emission spectra were of fine structure and contained two bands with emission maxima related to both local excitations (LE, λ_max_ = 416 (**7e**), λ_max_ = 420 nm (**8e**)) and the state of intramolecular charge transfer (ICT, λ_max_ = 520 (**7e**), λ_max_ = 524 nm (**8e**)). To explain this phenomenon, a more detailed study of the effect of solvent polarity on the nature of the excited state of fluorophores **7e**–**8e** was required.

A significant hypsochromic shift of the emission maxima (λ_em_ = 421 nm) was observed only for the cycloheptene-fused pyranoindole **7h**, which has the least energetically favorable state of the resulting series of fluorophores, ΔE = 6.29 eV (Appendix A).

Fluorophores **7**–**8** demonstrated high Stokes shifts (up to 14,682 cm^−1^) along with high quantum yields (up to 76%) in the MeCN medium.

#### 2.2.2. Influence of Chemical Functionalization on the Photophysical Properties

We evaluated the effect of the substituents in compounds **7** and **8** at positions 3, 4, and 8, 9 on the photophysical characteristics. It was found that the highest quantum yields with sufficiently high values of the Stokes shift were shown by compounds **7c** and **8c** containing an n-propyl substituent at positions 4 and 9, respectively. Replacement of the phenyl moieties in compound **7** with 3,4,5-trimethoxyphenyl residues resulted in an almost complete disappearance of the luminescence of compound **12**, with a bathochromic shift in the absorption and emission maxima. These facts correlate with our previously founded pattern of the photophysical properties of 4-phenylcoumarins having a pyridyl substituent at position 8 [28]. Namely, we observed an almost complete absence of the luminescent properties for the compounds with the 4-phenylcoumarin scaffold.

Compared to compound **8g**, for compound **10** with an aromatized cyclohexene fragment, a bathochromic shift of the emission maxima by 41 nm was observed, but with a significant decrease in the quantum yield to 6% and a decrease in the Stokes shift by 384 cm^−1^. Thus, cyclopentene, cyclohexene, or cycloheptene-fused fluorophores, such as **2**, **3**, **7f**, **8f**, **7g**, **8g**, **7h**, and **9b**, exhibited shorter wavelength emission in the range from 416 to 490 nm with Stokes shift values not exceeding 12,060 cm^−1^ (with the exception of **7f** and **8f**).

The photophysical characteristics of pyrano[2,3-*f*]indoles **2** and **3** as well as pyrano[2,3-e]indoles **6a** and **6b** were also studied. Thus, compound **6** showed emissions in acetonitrile solutions during photoexcitation in the region of 474–522 nm with a quantum yield of up to 7% (Table 1). We also compared their properties with their isomers **7c**, **7i** and **8c**. Compound **6a** as compared to its isomer **7i** demonstrated a considerably higher value of Stokes shift (15,931 cm^−1^ vs. 12,680 cm^−1^), and this value was the maximum in the series of compounds studied.

Thus, most of compounds **2**, **7**, and **8** were characterized by high quantum yields and large Stokes shifts. In addition, their emission waves lie in a wide range from blue to green (Figure 5).

The photophysical characteristics of 6-methoxy-2,3-diphenyl-1*H*-indole **13** as well as mono- and dimethoxy-containing 4-propyl-2*H*-chromen-2-one **14** and **15** were studied (Table 1). The results show that, almost in all cases, combining these two luminophores into a single system lead to a marked bathochromic shift in the absorption and emission maxima as well as a marked increase in the Stokes shift values. The only exceptions were compounds **7h** and **7g** (Table 1) mentioned above. Thus, for compounds **13**–**15**, the longest wavelength emission maxima was 418 nm, and the maximum Stokes shift value was only 7036 cm^−1^. For example, the prospects for combining them into a single chromophoric system, performed within the framework of this work, are obvious in terms of optimization of all photophysical parameters.

N-Alkylation of compounds **7** (see Table 1, compounds **9a**–**d**) in all the cases resulted in a noticable decrease in luminescence quantum yields, especially for compounds **9c** and **9d**, with a significant red shift to the long wavelength region up to 586 nm with a Stokes shift of 10,694 cm^−1^.

In general, we should note that the majority of the compounds studied, particularly, angular pyranoindols **8**, had significantly high Stokes shifts (up to 15,000 cm^−1^) and high quantum yields (up to 66%) achieved by a proper combination of the chromen-2-one and 1*H*-indole chromophores into a single planar pyranoindole π-conjugated system with the possibility of tuning the photophysical properties via the degree of the conjugation angle/length and the nature of functional groups including the cyclopentene, cyclohexene, or cycloheptene fragments.

#### 2.2.3. Solvent Effect and Intramolecular Charge Transfer

To elucidate the influence of the nature of intramolecular charge transfer in an excited state and the influence of the nature of solvents on the fluorescent behavior of chromophores, additional DFT calculations were carried out in combination with Lippert-Mataga mathematical models for the estimation of the general effect of solvents for fluorophores **7**–**8**. According to the frontier orbital DFT calculations and electrostatic potential visualization, isomeric pyranoindoles **7**–**9** represent a donor-acceptor model (D-A) consisting of a biphenyl-substituted 1*H*-indole electron-donor fragment (D) and a covalently bonded or fused electron-withdrawing 2*H*-pyran-2-one ring (A), and the polarity of the solvent can significantly affect the properties of the electronic states of the chromophore (Figure 6) [31,32].

As a next step, the emission of fluorophores **2**, **6**, **7**, **8**, **9**, **10**, and **12** was conducted, depending on the nature of the solvent, including the polarity and nature of hydrogen bonds. Indeed, the overall effect of the solvent polarity on the photophysical properties was observed for the entire range of chromophores (Appendix A). The difference between ground and excited dipole moments (Δμ) for the samples was calculated as the tangent of the slope of the Lippert-Mataga plot (ESI, formula 1). However, only the high linearity of the plots (R^2^ > 0.90) based on the linear Lippert-Mataga correlation equation [25] was significant evidence of a positive solvatochromic effect for samples **8c**, **8g**, and **9a** (Appendix A, Table 2).

Normalized plots of the emission in solvents of different nature and a single plot of the dependence of the Stokes shift of fluorophores **8c**, **8g**, and **9a** on the orientational polarizability of solvents Δf are shown in Figure 7a–d.

The emission spectra in a nonpolar medium (Δf = 0.0001 for *n*-heptane) revealed the predominance of the LE state for the studied chromophores, as evidenced by the fine structure of the emission spectra and low Stokes shift (no more than 90 nm) (ESI). In a polar medium at Δf > 0.2, the manifestation of the ICT state (440–550 nm) was confirmed by the presence of unstructured emission spectra bands with a Stokes shift of more than 200 nm (ESI). Replacement of aprotic solvents with protic, highly polar methanol induced the appearance of an LE-ICT hybrid state for derivatives **8c** and **9a** (Figure 7). For a more detailed study of the LE-ICT hybrid process, time-resolved fluorescence decay behavior was studied for **8c** and **9a** vs. **8g** with a pronounced ICT state in methanol. According to the obtained results, in pure methanol, the time-resolved fluorescence decay curves for **8c** and **9a** were bi-exponential (Appendix A), while those for **8g** were mono-exponential.

At the same time, the average fluorescence lifetime (τ_av_) for **8g** in methanol was 6.07 ns, and it was significantly higher than those for **8c** and **9a** (Appendix A). On the other hand, upon excitation at the maximum of the ICT band, the average lifetime was significantly higher (4.18 ns for **8c** and 1.64 ns for **9a**) than upon excitation at the maximum of the LE band (2.86 ns for **8c** and 1.15 ns for **9a**) (Appendix A).

Thus, a high degree of ICT in the excited state was observed for pyranoindole fluorophores, for example, Δμ ~ 15D for **8c** and **8g**. In particular, the observed Stokes shift values for **8c** were in a range from 7427 cm^−1^ (n-heptane) to 14,280 cm^−1^ (MeOH) (Appendix A). The experimental results obtained are in good agreement with the DFT calculations. Thus, the calculated values of the difference in the dipole moment of the excited state and ground state in the gas phase (Δμ) for the samples **8c** and **8g** were 9.3D and 8.1D, respectively, which correspond to the largest dipole moments calculated using the Lippert-Mataga model (15.9D and 15.1D, Table 2).

#### 2.2.4. Dual Fluorescence and Solvatochromic Study on Probes **7e**–**8e**

The fluorescence spectra of the compounds **7e**–**8e** were measured in a range of solvents spanning a scale of orientational polarizability from cyclohexane to MeOH (Appendix A). In low-polarity aprotic solvents, the fluorescence spectra of the samples consisted of single LE bands, which shifted to the red region upon increasing solvent polarity. When the solvent polarity exceeded the threshold value of the orientational polarizability, such as Δf > 0.2, the second band of the ICT process appeared, which was emitted with a further increase in the solvent polarity, including protic methanol. Thus, starting from THF, a dual (hybrid) fluorescence was observed in all solvents in the wavelength ranges of 390–475 nm and 480–740 nm, which corresponds to the generally accepted mechanism of the excited state with extended conjugation (ESEC) [33].

### 2.3. Theoretical Calculation

In order to correlate the photophysical properties of the pyranoindole fluorophores **7**–**9** with their structures, all theoretical calculations in gas phase were performed using density functional theory (DFT) with Gaussian 09 quantum chemistry software at the CAM-B3LYP/6-31+G* level of theory [34]. The highest occupied molecular orbital (HOMO) and the lowest unoccupied molecular orbital (LUMO) frontier orbital maps for compounds **7a**, **7c**, **7d**, **7e**, **7g**, **8a**, **8c**, **8d**, **8e**, **8g**, and **9a** can be found in Appendix A.

#### 2.3.1. Model of “Push-Pull” Pyranoindole Fluorophores

Based on DFT calculations, the analysis of the electron density distribution of HOMO-LUMO in the ground state in the gas phase was carried out along with the data of visualization of the electrostatic distribution and energy gap values, in order to construct the most realistic donor-acceptor models for samples **7**–**9**. Thus, at the energy levels of HOMO fluorophores **7**–**8**, the orbital of the donor biphenyl-containing indole fragment (blue color) predominated. Whereas in LUMO orbitals, the acceptor pyran-2-one group (red) was the main contributor (Figure 6).

Additional evidence supporting this model also comes from both electrostatic potential imaging data (Appendix A) and additional HOMO-1 and LUMO+1 calculations for compounds **7c**–**8c** (Appendix A). Indeed, the electron density of HOMO-1 and LUMO+1 **7c**–**8c** was distributed even more locally than in the HOMO-LUMO orbitals, which suggests a significant "push-pull" character of the herein reported pyranoindole fluorophores. This correlates well with the experimental data on the study of solvatochromism, based on which the maximum difference in dipole moments in the ground and excited states was from 11 to 15D for samples **7**–**9** according to the Lippert-Mataga model (Δμ > 10D).

#### 2.3.2. Structure–Property Relationships for the ICT-State

Based on the analysis of the above data of photophysical studies and theoretical calculations, it can be argued that tuning the properties of the ICT state of a number of pyranoindole fluorophores **7**–**9** of the D-A type is possible both by varying the nature of the functional groups in positions 3-4 of the pyranoindole domain for the series **7**,**9** and **8**–**9** for series **8**, and by changing the electron transfer angle for two isomeric pyranoindole series with linear [3,2-*f*] (**7**,**9**) and angular [2,3-*g*] (**8**) type of ring fusion. Thus, a significant redshift of the CT-band in acetonitrile (520 nm and 524 nm) was observed only in 3- and 8-chloro-substituted compounds **7e**–**8e**, which have the most energetically favorable states of the resulting series of fluorophores (6.19 eV for **7e** and 6.15 eV for **8e**, the lowest values of ΔE in the gas phase, Table 3). These facts can be explained by improving the conjugation of the chromophore model, on the one hand, due to the electron-donor mesomeric effect of the chlorine atom, and, on the other hand, by increasing the acceptor nature of the 2*H*-pyran-2-one ring due to the electron-withdrawing inductive effect of the chlorine atom. However, the experimental values of the dipole moments were lower than the theoretically calculated ones (Appendix A and Table 3) due to the competing delocalization of electrons due to the influence of the chlorine atom, and this resulted in the generation of many species with separated charges [30].

The subsequent replacement of 3- and 8-chloro with benzyl/alkyl and cycloalkyl functionalization led to a noticeable hypsochromic shift in the emission spectra and, resulted in a decrease in the energetically favorable state and an increase in the energy gap in the gas phase (Table 3, Figure 8).

Based on the electron density distribution of frontier orbitals and the difference in the dipole moment of the excited state and the ground state, it could be assumed that the degree of intramolecular charge transfer is higher for angular pyranoindoles **8** compared to linear ones **7**. Nevertheless, even regions of the molecule that do not present a net change in total density can show an alternation of excess and depletion of density as a consequence of electronic transition, thus making it difficult to directly use the density distribution maps to follow and quantify the charge-transfer (CT) phenomena [35].

The additionally calculated index (D_CT_) of the spatial extension of the ICT transition for linear isomers **7a**–**c** turned out to be lower than that for angular isomers **8a**–**c** (for example, for **7a** D_CT_ = 1.388 Å, for **8a** D_CT_ = 1.446 Å, Appendix A), which is convincing evidence for a higher degree of intramolecular charge transfer observed in the case of angular isomers **8**.

#### 2.3.3. Structure–Property Relationships for the Stokes Shift

According to the data in Table 1, there are significant differences between the Stokes shifts of isomeric pyranoindoles **7**,**9** and **8** with linear [3,2-*f*] and angular [2,3-*g*] type of ring fusion. These differences cannot be explained by the influence of the solvent alone. Therefore, the differences in the Stokes shift between linear and angular pyranoindoles are closely related to the structural features of the isomers and the nature of the substituents. Thus, the largest Stokes shift was observed for 3- and 8-chloro-substituted compounds **7e**–**8e**. In particular, the resonance effect, which causes molecules to have a tendency to align with their molecular plane to achieve better electron delocalization, explains the high Stokes shifts for linear isomer **7e** (Figure 9) [36]. The replacement of chlorine atoms by benzyl/alkyl and cycloalkyl functionalization led to a decrease in the Stokes shift (Figure 10).

For each pair of linear [3,2-*f*] and angular [2,3-*g*] isomers of pyranoindoles **7**–**8** with the same substituents it would be reasonable to predict that the higher electron density delocalization of FMO provides the greater nuclear motion in the relaxation time of the geometry of the excited state. Therefore, the HOMO-LUMO overlap index, Λ, gives a good indication of the Stokes shift values, and the smaller Λ correlates well with the larger Δλ [36]. Indeed, the calculated index of boundary orbital overlap Λ for pyranoindole **7b** was smaller (0.76332 a.u.), hence the Stokes shift was larger than for **8b** (0.78534 a.u.).

## 3. Conclusions

In conclusion, we designed and synthesized a number of new pyrrolo-coumarin fluorescent dyes. Bischler–Möhlau reaction of 3-aminophenol with benzoin followed by Pechmann condensation of the formed hydroxyindoles with β-ketoesters provides a straightforward pathway to pyrano[3,2-*f*], [2,3-*g*], and [2,3-*e*]indoles, whereas pyrano[2,3-*f*]indoles were obtained via the Nenitzescu reaction of p-benzoquinone and ethyl aminocrotonates and subsequent Pechmann condensation of 5-hydroxyindole derivatives. The use of inexpensive ketoesters, aminophenol, and benzoquinone as the common starting materials was an additional advantage of this study, providing an economic route to future applications of pyranoindole dyes.

The results of photophysical studies and DFT calculations showed that isomeric pyranoindoles **7**–**9** as donor-acceptor (D-A) chromophores could generate photoinduced intramolecular charge-transfer states between biphenyl-substituted 1*H*-indole electron-donor fragment (D) and covalently bonded or fused electron-withdrawing (A) 2*H*-pyran-2-one moiety. For the isomeric pyranoindoles, general photophysical properties such as absorption, emission, lifetime, and QY in solution were studied. The emission of fluorophores **7**–**9** covered the range from blue to green, with emission maxima from 420 to 586 nm.

In contrast to previous work [15], wherein compounds characterized by quantum yields of 0.20–0.57 and Stokes shifts of 5652–6588 cm^−1^ were prepared, we herein found leading compounds with high quantum yields up to 0.89 and very large Stokes shifts about 14,000 cm^−1^ (Figure 11). Furthermore, our proposed method requires easily available starting materials.

It was concluded that, except for compounds **7e**–**8e**, all the studied compounds exhibited the ICT character of the excited state, which was confirmed by means of solvatochromic experiments and theoretical calculations. In solvents with high orientational polarizability (Δf > 0.2) 3- and 8-chloro-substituted samples exhibited dual fluorescence.

Based on the FMO electron density distribution and dipole moment values, it was suggested that the degree of ICT processes for angular pyranoindoles **8** was higher than linear **7** with similar substituents, and it was additionally confirmed by the calculation of the index (D_CT_) of the spatial extension of the ICT transition. The calculation of the frontier orbital overlap index Λ, together with the resonance effect, made it possible to explain the higher Stokes shifts for linear isomers compared to angular ones.

A general correlation was established for the influence of the nature of substituents at positions 3-4 of the pyranoindole domain on the degree of the ICT state and the Stokes shift for series **7**,**9** and **8**,**9**, such as cycloalkane < Alk, Bn < Cl.

Understanding the relationships between fluorophore structures and their photophysical properties makes it possible to fine-tune the optoelectronic properties of dyes by means of proper synthetic modifications of the fluorophores. We believe that the realization of such fundamental knowledge will be very useful for rational molecular engineering in order to obtain more efficient pyranoindole dyes and samples. In particular, the ability of 3- or 8-chloro-containing pyranoindoles to undergo a significant charge redistribution depending on the polarity of the solvent in the excited state, or, in other words, to change their photophysical properties under the influence of the local microenvironment, in combination with the ability to switch between single and double luminescence, suggests their potential application as polarity probes in various microenvironments.

## Data Availability

Not applicable.

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
