# Peer review of "Fluorescent Pyranoindole Congeners: Synthesis and Photophysical Properties of Pyrano[3,2-f], [2,3-g], [2,3-f], and [2,3-e]Indoles"

_molecules, 2022, doi:10.3390/molecules27248867_

Round 1
Reviewer 1 Report
Ainur D. Sharapov et al reports the synthesis of four types of annulated pyranoindole derivatives (pyrano[3,2-f]indole, pyrano[2,3-g]indole, pyrano[2,3-f]indole, and pyrano[2,3-e]indole) and studied their photophysical properties. Among the pyranoindoles studied, the pyrano[3,2-f] and [2,3-g]indoles are showed good properties including moderate to high quantum yields (30-89%) and large (9000-15000 cm-1) Stokes shift. Authors synthesized various analog by starting from different acetoacetic esters and differently substituted indole derivatives. Also tested the N-alkyl analogs. Best analogs were further validated by testing in different solvent systems and performed the quantum chemical calculations.
Overall, the work reported in this is novel and studied in detail. I recommend publishing after addressing the following minor queries.
1. The discussion too lengthy, it needs to be to revise to be concise
2. Table can be replaced by showing the R1, R2 in the scheme itself. Table can go to SI.
3. I recommend testing one N-aryl analog
4.
Reviewer 2 Report
In section 2.1.2, kindly mention the solvent and ratio used to separate compounds 4 & 5.
Why do authors choose only acetonitrile to study the photophysical properties of compounds?
Is the author predicted only in the ground state? What is the significance behind it? In addition, the author should compare the dipole moments of compounds in the ground and excited states and indicate whether both dipole moments are parallel.
It would be better to compare their findings to those in the literature.
In Figure 7, why the background at longer wavelengths is too high in solvents? Are there any solubility issues?
The conditions of the measurements must be presented in figure caption 7 (slits, excitation wavelength).
Reviewer 3 Report
|
The authors describe the synthesis of various types of pyranoindoles with different ring fusion and organic groups and provide an in-depth study of their photophysical properties.
The manuscript is well-planned, and its simple structure/timeliness makes it very easy to read and understand. The discussion is complete and accurately elaborated, describing all the synthesised molecules and the techniques used to characterise all the derivatives (RMN and X-ray diffraction), as well as a comprehensive in-depth study of the photophysical properties, with only some minor typos throughout the manuscript. However, the results here presented merit some comments and arise some unresolved questions.
- It is not clear why 7c,g,i have been chosen for alkylation. Needs clarification in the text (line 140)
- Are any of the pyranoindoles soluble in water? And if so, to what extent? A comment on the solubility in water of the compounds should be added as the study of fluorescence decay in water would be interesting for these compounds.
- Most of the complexes that exhibit high quantum yields and high Stokes shifts are the angular version (8a, 8c, 8d, 8f, 8g, except for 7c). A plausible reason for this observation should be considered and added to the manuscript’s main text.
- Minor grammatical English typos throughout the manuscript. Extensive proofreading needs to be performed by the authors:
· Line 93. The overall yield is 78%. Needs correction.
· Line 109. Photophysical properties are not shown in Scheme 3.
· Lines 187-195 The paragraph is out of context. The font size is smaller than it should be, and the starting line is not understandable (Table 275…). The same occurs in lines 199-201. I think the location is mistaken. Please relocate. Something similar is observed in lines 259-260 (the sentence should be rephrased to better understand the reader).
· Line 253, compound 2 should be deleted as it does not exhibit a high quantum yield.
· Line 297. “The effect of the overall effect”.. should be corrected to “the overall effect…”
· And so on… |
